# Preparation and Properties of (Sc_2_O_3_-MgO)/Pcl/Pvp Electrospun Nanofiber Membranes for the Inhibition of *Escherichia coli* Infections

**DOI:** 10.3390/ijms24087649

**Published:** 2023-04-21

**Authors:** Yanjing Liu, Xiyue Li, Yuezhou Liu, Yaping Huang, Fuming Wang, Yongfang Qian, Ying Wang

**Affiliations:** School of Textile and Material Engineering, Dalian Polytechnic University, Dalian 116034, China

**Keywords:** nano-textured, Sc_2_O_3_-MgO, antibacterial, electrospinning

## Abstract

Due to their high porosity, large specific surface area, and structural similarity with the extracellular matrix (ECM), electrospun nanofiber membranes are often endowed with the antibacterial properties for biomedical applications. The purpose of this study was to synthesize nano-structured Sc2O3-MgO by doping Sc^3+^, calcining at 600 °C, and then loading it onto the PCL/PVP substrates with electrospinning technology with the aim of developing new efficient antibacterial nanofiber membranes for tissue engineering. A scanning electron microscope (SEM) and energy dispersive X-ray spectrometer (EDS) were used to study the morphology of all formulations and analyze the types and contents of the elements, and an X-ray diffraction (XRD), thermogravimetric analysis (TGA), and Fourier transform attenuated total reflection infrared spectroscopy (ATR-FTIR) were used for further analysis. The experimental results showed that the PCL/PVP (SMCV-2.0) nanofibers loaded with 2.0 wt% Sc_2_O_3_-MgO were smooth and homogeneous with an average diameter of 252.6 nm; the antibacterial test indicated that a low load concentration of 2.0 wt% Sc2O3-MgO in PCL/PVP (SMCV-2.0) showed a 100% antibacterial rate against *Escherichia coli* (*E. coli*).

## 1. Introduction

A wound dressing is a covering or protective layer that can temporarily protect the damaged skin in the process of wound healing and treatment and avoid or control wound infection [1]. Therefore, it is very important to develop wound dressing that can prevent bacteria from penetrating into the wound or avoid microbial growth. Over the past few decades, extensive research has been conducted on wound dressings with antibacterial properties, such as thin films, hydrogels, emulsions, composites, nano/microfibers, etc. [2]. In recent years, nano/microfibers have shown broad application prospects in nano-wound dressing. The electrospun nanofiber structure possesses absorbability, bacterial barrier, oxygen permeability (gas transfer), non-adhesion to healing tissue, and biological activity, all of which are essential properties for antibacterial wound dressings [3]. Nanofiber membranes are widely used in the fields of biomedical applications and tissue engineering due to the unique characteristics, including tunable porosity, large specific surface area, high aspect ratio, controllable small pore size, and the ability to resemble the architecture of an extracellular matrix [4,5,6]. Among the available techniques, electrospinning is one of the most promising methods of manufacturing nanofiber membranes in tissue engineering applications [7]. Electrospinning is an effective nanofiber fabrication process that stretches unfragmented polymer fibers from a polymer solution or polymer melt with electrostatic force using a high voltage direct current in the form of a liquid jet preparation of nanofibers [8,9]. As ideal wound dressings, electrospun nanofiber membranes have a similar structure to an extracellular matrix (ECM), which can promote the interaction between cells at the wound surface. Owing to the advantages of a unique uniform and dense network structure, large specific surface area, strong flexibility, and high porosity, electrospun nanofiber membranes can be used as drug carriers to the wound, which have great potential in the development of antibacterial dressings. 

Currently, nanofibers of natural and synthetic polymers, such as gelatin (GEL), chitosan (CS), polycaprolactone (PCL), and polyvinylpyrrolidone (PVP), are successfully prepared by electrospinning as biomimetic and temporal substrates to regulate cellular and molecular activities [10,11,12,13]. PCL is a biodegradable semi-crystalline polymer with excellent mechanical properties, it cannot be dissolved in water, and it is easily dissolved in a variety of polar organic solvents. PCL is characterized by good biocompatibility, non-toxicity, adjustable degradation rate, permeability with many drugs, and complete absorption and metabolism from the human body, which is widely used in controlled drug release systems as a sustained release carrier material [14]. It is commonly used in surgical sutures, fracture internal fixation devices, drug delivery, and tissue or organ regeneration scaffolds [15]. However, PCL is a hydrophobic material with strong crystallization and poor hydrophilicity, which hinders its application in medical fields where a rapid absorption rate is needed [16]. Its slow degradation kinetics may hinder its application in some biomedical applications that require faster absorption rates. The crystallinity of PCL can be reduced by mixing it with other polymers, which can result in a suitable hydrophilicity and can be conducive to cell adhesion. Polyvinylpyrrolidone (PVP) is a biodegradable, biocompatible, water soluble, ph stable, non-toxic amphiphilic polymer with good solubility, viscosity, and film-forming performance with a variety of organic solvents. PVP has good electrostatic spinning properties, including spinning and fiber extraction, and is widely used in the preparation of nanocapsules, implant materials, and scaffolds [17]. PVP can be used to produce materials with adjustable fiber surface morphology and degradation rate so that the hydrophilicity and biodegradability of PCL nanofibers can be improved by adding PVP for blending electrospinning [18,19]. PCL and PVP have shown excellent biocompatibility and effectiveness both in vivo and in vitro and have no irritation to the skin, mucous membranes, and eyes as wound dressings. 

As an environmentally friendly inorganic nanomaterial, nano magnesium oxide (nano-MgO) has good biocompatibility, durable and broad-spectrum antibacterial activity, and the ability to avoid other antibacterial materials that are expensive, biotoxic, and prone to discoloration failure and light dependence, so it has become a kind of inorganic antibacterial material with great application potential. Nano-MgO shows unique advantages in good bactericidal and inhibitory ability on bacteria, fungi, and cancer cells [20,21,22,23], which can effectively destroy cell membrane structures and kill bacteria through the oxidative damage of reactive oxygen species (ROS) and mechanical damage caused by adsorption. Organic antimicrobials have strong initial bactericidal power, but their chemical stability is poor, and they are easy to volatilize when exposed to heat, light, or water, so it is difficult to achieve a long-lasting effect, and they even produce toxic decomposition products. The smaller the particle size of the nanometer magnesium oxide, the better the growth inhibition and destruction of the bacteria [24]. In addition, ROS play an important role in the antibacterial mechanism of magnesium oxide. This depends mainly on the oxygen vacancy and alkalinity of magnesium oxide. Ion doping is an effective method to modify the physical and chemical properties of metal oxides, such as a small particle size, high defect concentration, and high catalytic activity. MgO nanoparticles have been incorporated into various polymer nanofibers to confer antibacterial properties [25]. Doping elements can cause lattice defects, such as vacancies, void atoms, replacement atoms, and dislocations, improving the antibacterial ability of pure magnesium oxide. Studies have shown that doping Zn, Cu [26], Ag [27], and ZnO nanoparticles [28] in nano-MgO can improve the physical and chemical properties of the material. Sc^3+^ can replace the original ionic lattice position and increase the defect material. Sc^3+^ can replace the original ionic lattice position and increase the defect concentration of the crystal, generating more ROS in the antibacterial process and improving the antibacterial performance of the material. Furthermore, magnesium (Mg), as one of the most abundant cations in the human body, is an essential element for human metabolism. Nigam et al. [29] used the MTT method to test the biocompatibility of MgO nanoparticles, and the results showed that MgO was compatible with human cells and safe for human use. Vijayakumar et al. [30] and Tabrez et al. [31] respectively evaluated the cytotoxicity of MgO NPs in human ovarian teratoma (PA-1) cells and found good biocompatibility and stability. Li et al. [32] studied the effect of scandium on the biocompatibility of degradable magnesium alloys. With low concentration Sc^3+^ doping on L-929 and SP2/0 cell MTT detection, the cell survival rate was more than 70% of the negative control—no cytotoxicity. Sc^3+^ doping has an acceptable biosafety in cell metabolism. In summary, Sc_2_O_3_-MgO has better cell activity and biocompatibility and no cytotoxicity when loaded on PCL/PVP nanofibers.

In this work, the purpose of this paper is to obtain electrospun nanofiber membranes (Sc_2_O_3_-MgO)/PCL/PVP (SMCV) with excellent antibacterial properties that have potential application in the field of medical wound dressing. By electrospinning technology, Sc_2_O_3_-MgO was mixed into a spinning solution to prepare a PCL/PVP nanofiber membrane, which had antibacterial properties. Electrospun fiber membranes loaded with bioactive materials can be used for wound dressing, drug release, and artificial tissue engineering scaffolds. Loaded nanoparticles physically protect wounds from bacterial activity. Nanofibers can help cell differentiation and proliferation. The Sc_2_O_3_-MgO with a nano-textured surface was an antibacterial agent, and PCL/PVP were used as nanofiber membrane substrates. The structures of SMCV were characterized by scanning electron microscopy (SEM), energy dispersive spectroscopy (EDS), infrared spectroscopy (ATR-FTIR) thermogravimetric analysis (TGA), and differential scanning calorimetry (DSC). The antibacterial performance of SMCV against *Escherichia coli* (*E. coli*) was tested with a modified shake-flask method.

## 2. Results and Discussion

### 2.1. Morphology and Diameter Analysis

In order to simulate the structure and function of the ECM, the tissue engineering structure must be conducive to promoting cell adhesion and proliferation. Scanning electron microscopy was used to observe the structure of nanofiber scaffolds, and the influence of the change in the Sc_2_O_3_-MgO load on the morphology of nanofibers was studied. The effects of Sc_2_O_3_-MgO loading on the morphology of the nanofibers were investigated by SEM. Figure 1 shows the SEM of SMCV electrospun nanofiber membranes loaded with different content of nano-textured Sc_2_O_3_-MgO (0, 0.5, 1.0, 1.5, 2.0, 2.5, and 3.0 wt%) and the corresponding fiber diameter distribution. As shown in SEM images, the Sc_2_O_3_-MgO is almost insoluble in spinning solutions due to its poor water solubility so that most of the antibacterial agents exist in the insoluble suspended particles in spinning solutions. The SEM images show some white granular materials with a nano size on the nanofibers, which are the Sc_2_O_3_-MgO that exist in PVL/PVP substrates. As can be seen in the figure, the electrospun fibers are successfully prepared without any beads; all the nanofiber membranes consist of smooth and homogeneous fibers. With increasing Sc_2_O_3_-MgO loading, the white particulate matter on the nanofibers gradually increases and fiber diameters vary mostly between 100 nm and 400 nm, with the average diameter first increasing with the loading to a maximum of 272.48 nm at SMCV-1.5, and then gradually decreasing at SMCV-2.0. When the loading capacity of Sc_2_O_3_-MgO reaches 2.5 wt%, fewer fibers are received when nanofibers are prepared by electrostatic spinning technology due to the agglomeration of Sc_2_O_3_-MgO, and when it reaches 3.0 wt%, the agglomeration of nanoparticles increases and the particles are mostly solid particles. In this way, non-water-soluble Sc_2_O_3_-MgO will affect electrostatic spinning, reduce the spinning performance of PCL/PVP, and make the spinning process unstable, including affecting the continuity of spinning and the uniformity of nanofiber morphology and diameter.

### 2.2. EDS Spectrum

EDS characterization was used to further determine the loading of Sc_2_O_3_-MgO and the distribution of elements in PCL/PVP nanofiber membranes, and the types and contents of SMCV components were analyzed. Figure 2a confirms the presence of carbon (C) at Ebinding = 0.3 keV, nitrogen (N) at Ebinding = 0.4 keV, and oxygen (O) at Ebinding = 0.5 keV. These observations show that the substrate of all seven fibers is a PCL/PVP component [33,34]. It is worth noting that the EDS spectra of SMCV (SMCV-0.5, SMCV-1, SMCV-1.5, SMCV-2.0, SMCV-2.5, and SMCV-3.0) confirm the presence of magnesium (Mg) in Ebinding = 1.3 keV and scandium (Sc) in Ebinding = 0.4 keV, Ebinding = 4.1 keV, and Ebinding = 4.5 keV. Thus, SMCV is mainly composed of elements C, N, O, Mg, and Sc, the results indicated that Sc_2_O_3_-MgO was loaded on the PCL/PVP substrate using electrospinning.

The results show that as the content of Sc_2_O_3_-MgO increases from 0.5 wt% to 3.0 wt% (SMCV-0.5, SMCV-1, SMCV-1.5, SMCV-2.0, SMCV-2.5 and SMCV-3.0), the calculation results are consistent with Table 1 and Figure 2a. EDS mapping is a direct method of reflecting the dispersion of elements in a sample. As shown in Figure 2b, yellow dots, red dots, dark green dots, light green dots, and blue dots are derived from elements C, O, N, Mg, and Sc, respectively. It can be clearly seen that the color darkness of C and O in SMCV begins to brighten with the increase in the content of Sc_2_O_3_-MgO, and gradually darkens when the load reaches 2 wt%. When the load of Sc_2_O_3_-MgO reaches 2.5 wt%, the light green and blue spots of Sc and Mg gradually darken. Therefore, the element mapping results show that Sc_2_O_3_-MgO is successfully loaded and evenly distributed in the SMCV fiber membranes. However, when the content of Sc_2_O_3_-MgO reached 2.0 wt%, the nanoparticles began to agglomerate obviously (Table 1).

### 2.3. XRD

The crystal state of SMCV loaded with an antibacterial agent was analyzed with an XRD test, and the corresponding diffraction curve was analyzed and studied. The crystalline structures of fibers SMCV (SMCV-0, SMCV-0.5, SMCV-1.0, SMCV-1.5, SMCV-2.0, SMCV-2.5, and SMCV-3.0) were tested with XRD in the range of 10~80°, and the results are shown in Figure 3a. As can be seen from the red vertical line in Figure 4a, the main diffraction indices are (111), (200), (220), (311), and (222), respectively. The corresponding diffraction peaks are 2θ = 36.863°, 42.825°, 62.169°, 78.445°, and 74.517°, indicating the polycrystalline properties of nano-MgO. After loading for Sc_2_O_3_-MgO, the nanofibers have sharper diffraction peaks, indicating that the crystallinity is improved after loading. In addition, in the green vertical line, it can be seen that the diffraction indices are (211), (222), (440), and (622). The corresponding diffraction peaks are 2θ = 22.178°, 31.567°, 52.741°, and 62.778°, which can be attributed to the load of Sc_2_O_3_-MgO. It shows that it exists in SMCV fiber samples. However, diffraction peaks belonging to Sc can be observed in all fiber felt of SMCV. This phenomenon indicates that the Sc is dispersed in a crystalline state in the fiber mat.

### 2.4. ATR Spectra Analysis

ATR was used to test whether new chemical bonds and molecular interactions were generated in SMCV nanofibers, and the functional groups of organic compounds were quickly and effectively identified. In ATR analysis through Figure 3b, it is shown that Sc_2_O_3_-MgO is successfully integrated into PCL/PVP electrospinning fibers. According to the resulting spectra (Figure 3b), the bands observed at 2945.07 cm^−1^ and 2986.59 cm^−1^ are attributable to the -CH_2_- and vibration of the –C=O stretch in the PCL spectrum. Compared with SMCV-0, bands at 3695 cm^−1^ showed the presence of Sc_2_O_3_-MgO nanoparticles. The SMCV spectra showed the dominant peaks of PCL (-C=O- stretching 1723.67 cm^−1^) and PVP (C=O stretching 1658.44 cm^−1^) [35]. According to the ATR-FTIR spectra of SMCV, the absorption bands are 1421.51 cm^−1^–1461.52 cm^−1^ (C-H deformation), 1366.74 cm^−1^ (C-O symmetric stretching), and 1236 cm^−1^ (C-O-C asymmetric stretching), respectively. In addition, two other characteristic peaks were found to be -O-C-O stretching at 1240.68 cm^−1^ for PCL and C-N stretching at 1289.96 cm^−1^ for PVP. In the spectrum of SMCV, the bands visible in the range of 652 cm^−1^ to 400 cm^−1^ belong to MgO vibrations [36]. With an increase in the Sc^3+^ content, the obvious bands that are less than 800 cm^−1^ belong to the metal oxygen bond, which proves that Sc_2_O_3_-MgO is successfully loaded into PCL/PVP electrospinning fibers. It also proves that the increase in the load of Sc_2_O_3_-MgO leads to a slight deviation in the peak position of the infrared spectrum but to no obvious change in the chemical structure.

### 2.5. Thermal Analysis of Nanofibers

Thermogravimetric analyzer (TG) was used to test the thermodynamic properties of SMCV nanofibers and characterize their thermal stability. As shown in Figure 4c, the thermal decomposition behavior of the electrospun fiber samples of SMCV was characterized by TG at a nitrogen flow rate of 25 mL/min increased from 30 °C/min to 800 °C. The weight loss properties of water evaporation and thermal decomposition of SMCV fibers were calculated as a function of temperature. The first stage of weight loss is due to evaporation. The major weight loss in the second stage between 250 °C and 350 °C was due to normal thermal decomposition of PVP and PCL. SMCV-0 starts to lose weight at 250 °C, and it loses weight completely at 355 °C. Compared with SMCV-0, the thermogravimetric temperature of SMCV after loading Sc_2_O_3_-MgO is delayed by 10 °C. As a result, the thermal stability of SMCV-0 is significantly higher than that of original SMCV-0, and the residual mass is also higher than that of unloaded nanofiber. The final weightlessness of more than 460 °C is due to the elimination of carbon residues and nanoparticles produced by thermal reactions. The third phase of the weight variation of SMCV indicates that the addition of Sc_2_O_3_-MgO can improve the thermal performance of SMCV.

### 2.6. XPS Analysis of Nanofibers

XPS was used to test the SMCV nanofiber membrane before and after loading the Sc_2_O_3_-MgO, and the content changes of elements on the surface of the membrane were determined. Before and after loading Sc_2_O_3_-MgO onto SMCV nanofiber membrane with XPS, the content of elements on the membrane surface was measured. Figure 5a,b is the full-scan spectrum analysis of SMCV-0 and SMCV-2.0 surface element content, showing that the test samples mainly contain characteristic peaks of C, O, and N. In the figure, there is no impurity peak of other substances, and the content of C element and O element in the nanofiber membrane is above 95%, which is the main component of the PCL/PVP spinning substrate. As shown in Figure 5c,d, the increases of 1303.3 eV and 398.4 eV are mainly attributed to the characteristic peaks of Mg2p and Sc2p; the doping of Sc^3+^ may change the molecular structure of MgO, in which the increase of Sc^3+^ is more obvious, indicating that Sc_2_O_3_-MgO is successfully loaded on SMCV. 

### 2.7. Antibacterial Activity of Nanofibers

The number of *E. coli* colonies in the LB solid medium of SMCV is recorded, and the calculation results are shown in Table 2. The number of *E. coli* colonies on LB solid medium was 162 ± 14. The number of *E. coli* colonies in control group, SMCV-0, SMCV-0.5, SMCV-1.0, SMCV-1.5, SMCV-2.0, SMCV-2.5, and SMCV-3.0 were 182.0 ± 14.1, 173.0 ± 14.5, 141.6 ± 8.1, 136.0 ± 9.2, 45.3 ± 10.5, 0 ± 0, 1.0 ± 1.2, and 27.5 ± 8.1, respectively. The results showed that SMCV (SMCV-0, SMCV-0.5, SMCV-1.0, SMCV-1.5, SMCV-2.0, SMCV-2.5, and SMCV-3.0) on LB solid medium decreased with increasing Sc_2_O_3_-MgO content in SMCV. The number of *E. coli* colonies in SMCV was dose-dependent with loaded antibacterial Sc_2_O_3_-MgO. The percentage reduction in the number of *E. coli* colonies was calculated with equation (1). As shown in Figure 4b, SMCV has certain antibacterial abilities against *E. coli*, and the antibacterial rate increases significantly with the increase in loading capacity while the antibacterial performance weakens when the load is too large (18.1 ± 1.19, 31.32 ± 2.52, 61.50 ± 3.03, 100 ± 0, 93.33 ± 1.15, 83.01 ± 2.93%, *p* < 0.05 or *p* < 0.01).

The antibacterial property of SMCV loaded with different weights of nano-textured Sc_2_O_3_-MgO (0, 0.5, 1.0, 1.5, 2.0, 2.5, and 3.0 wt%) against *E. coli* were tested by the modified flask standard method. After 24 h of culture, *E. coli* colonies are observed on LB solid medium, as shown in Figure 5a. It can be clearly seen that the control LB solid medium is covered with *E. coli* colonies while the number of *E. coli* colonies was slightly reduced on the LB solid medium of SMCV-0.5. When the content of Sc_2_O_3_-MgO reached 2.0 wt%, there are almost no *E. coli* colonies on the LB solid medium of SMCV-2.0, and the antibacterial rate reaches 100%. Therefore, with the increase in Sc_2_O_3_-MgO content in SMCV, the number of *E. coli* colonies on LB solid medium in SMCV gradually decreases while aggregation occurs when the load reaches 2.5 and 3.0 wt%, limiting the dispersion of antibacterial agent. *E. coli* colonies grow on LB solid medium of SMCV-2.5 and SMCV-3.0. These results indicate that SMCV has a certain bacteriostatic rate against *E. coli*, and the antibacterial rate can be increased with an increase in Sc_2_O_3_-MgO content in SMCV. Furthermore, a 2.0 wt% loading was the optimal antibacterial concentration.

In conclusion, the antibacterial performance of SMCV was completely dependent on the content of Sc_2_O_3_-MgO in SMCV, and the antibacterial performance was better with the increase in Sc_2_O_3_-MgO content. Low Sc_2_O_3_-MgO content in SMCV (0.5 wt%, 1.0 wt%) will result in poor antibacterial performance. However, due to the high content of Sc_2_O_3_-MgO in SMCV (2.5 wt%, 3.0 wt%), it is difficult for Sc_2_O_3_-MgO to be uniformly dispersed in SMCV, and the significant concentration of Sc_2_O_3_-MgO in SMCV may lead to poor availability. Therefore, SMCV-3.0 was prepared with the appropriate addition of 2.5 wt% Sc_2_O_3_-MgO, which gave SMCV both usable properties and antibacterial properties. Considering that the antibacterial properties and availability of SMCV have attracted much attention, an optimized content of 2.5 wt% Sc_2_O_3_-MgO was selected to prepare SMCV.

There are several theories about the antibacterial mechanism of MgO, but the production of reactive oxygen species (ROS) and mechanical damage are the main mechanisms. On the one hand, it relies on the morphological structure of magnesium oxide itself to physically damage the membrane system of bacteria, leading to cell rupture and death; on the other hand, the superoxide ions generated on its surface and the alkaline environment caused by the reaction with water can cause the denaturation of bacterial DNA and proteins [37,38,39]. As shown in Figure 6, Sc_2_O_3_-MgO has strong antibacterial activity. One possible mechanism for its bacterial virulence includes the release of O^2−^, hydroxyl radical OH, and severe damage to cellular components by reactive oxygen species (ROS) [27]. These free radicals can damage the cell structure through strong REDOX properties and prevent the normal reproduction of bacteria to achieve an antibacterial effect. On the other hand, there are many kinds of active sites on the surface of Sc_2_O_3_-MgO, such as lattice limited hydroxyl, free hydroxyl, and ion holes, which can be used as adsorption and surface reaction centers and cause mechanical damage. Scandium (Sc^3+^) has been shown to have the ability to inhibit the growth of *Klebsiella pneumoniae*, *E. coli*, and *Pseudomonas aeruginosa* [40,41,42]. The purpose of doping Sc^3+^ in the nano-MgO lattice is to increase the lattice defects and then increase the ROS content produced during the antibacterial process to enhance the antibacterial property of MgO. When Sc_2_O_3_-MgO is successfully loaded on PCL/PVP, it has good antibacterial performance.

## 3. Materials and Methods

### 3.1. Materials

Magnesium (Ⅱ) chloride hexahydrate (MgCl2·6H2O), scandium (Ⅲ) chloride hexahydrate (ScCl3·6H2O), and ammonium carbonate ((NH4)2CO3) purchased from Shanghai McLean Biochemical Technology Co., Ltd. (Shanghai, China) were used to prepare the nano-textured Sc2O3-MgO. 2-2-2-Trifluoroethanol (TFE, Analytical reagent) obtained from Aladdin Biochemical Technology Co., Ltd. (Shanghai, China) was used as spinning solvents. Polycaprolactone (PCL, 80,000) and polyvinyl pyrrolidone (PVP, K-30, 58,000) purchased from Baoqian Plastic Chemical Material Co., Ltd. (Hangzhou, China) and Aladdin Biochemical Technology Co., Ltd. (Shanghai, China) were used to prepare nanofibers. The bacteria used in this experiment was Escherichia coli (E. coli, ATCC 25,922, Shanghai Luwei Technology Co., Ltd. (Shanghai, China)). In addition, agar, tryptone, NaCl, and yeast extract powder purchased from Shanghai Maclean Biochemical Technology Co., Ltd. (Shanghai, China) were also used for the antibacterial tests. All reagents were analytically pure and were used without further processing or purification.

### 3.2. Synthesis of Nano-Textured Sc_2_O_3_-MgO

As in our previous study, the ScCl_3_·6H_2_O (0.0050 mol) and MgCl_2_·6H_2_O (0.0450 mol) were dissolved in deionized water (25 mL). Solutions containing Sc^3+^-Mg^2+^ were added to the (NH_4_)_2_CO_3_ solutions (0.002 mol/mL 25 mL) and stirred at 600 rpm at room temperature (HJ-4B) for 6 h. The solution was washed and filtered three times and dried in an oven (DHG-9248A) at 70 °C for 4 h to give the Sc_2_(CO_3_)_3_-MgCO_3_ precursor. Briefly, the Sc-doped MgO nanoparticles were prepared via a high-temperature calcination method: the Sc_2_(CO_3_)_3_-MgCO_3_ precursor was added in a muffle furnace (KSL-1400X-A3), which was heated from a room temperature of 25 °C to a target temperature of 600 °C within 90 min, and then, it was maintained at 600 °C for 180 min. The process was ended as the temperature dropped to 500 °C, and the nano-textured Sc_2_O_3_-MgO samples were obtained when the temperature dropped to 25 °C [43].

### 3.3. Fabrication of Nanofiber Membranes

#### 3.3.1. Spinning Solutions

Briefly, PCL and PVP were dissolved in TFE solvent in a 1:1 ratio, and the total polymer content was 12 wt% by weight. The spinning solutions were prepared by stirring at room temperature overnight until completely dissolved. Finally, 0.5 wt%, 1.0 wt%, 1.5 wt%, 2.0 wt%, 2.5 wt%, and 3.0 wt% Sc_2_O_3_-MgO were added to the PCL/PVP spinning solutions, respectively, and stirred at room temperature for 1 h; then, ultrasonic dispersions were performed to obtain Sc_2_O_3_-MgO/PCL/PVP spinning solutions.

#### 3.3.2. Electrospinning Process

Typically, the polymer solutions were allocated into a 10 mL plastic syringe with a feed rate of 0.5 mL/h at 20 kV, and the needle was 12 cm away from the aluminum foil collection plate. The electrospinning process was carried out at room temperature with relative humidity below 50% (HZ-03). The collected fibers were dried in a vacuum drying oven (DHG-9248A, 40 °C) so that the solvent in the fibers could be completely volatilized. The sample labels of the nanofiber membranes are shown in Table 3.

### 3.4. Characterizations

SEM (JEOL, Tokyo, Japan) was used to characterize the surface morphology of the nanofiber membranes. The diameter of 100 fibers was randomly obtained by Image J software. X-ray photoelectron spectroscopy (Thermo Fisher Scientific, Waltham, MA, USA) on a VG MultiLab 2000 X-ray photoelectron spectrometer using Al-Kα (hλ = 1486.6 eV) radiation as the excitation source, and the spectra were calibrated by the C 1s peak (284.8 eV). In addition, EDS (Oxford Instruments, Oxford, UK) was used to characterize the content and distribution of the elements. The XRD patterns were tested using XRD (Shimazu Corporation, Kyoto, Japan) at 40 kV and 30 mA with Cu Kα radiation (λ = 1.5406 A). XRD patterns were recorded over the diffraction angles ranging from 10° to 80° (2θ) at a scanning rate of 5° min^−1^. The thermal stability of the membranes was determined with thermogravimetric analysis (TGA). The thermal stability test adopted the synchronous thermal analysis instrument (Linseis STA). The samples were weighed to about 10 mg and placed in a sealed aluminum crucible for testing. At a nitrogen flow rate of 25 mL/min, a double heating and cooling cycle from 20 °C to 500 °C was conducted at a rate of 20 °C/min to evaluate thermal gravimetry. The nanofiber membranes were analyzed with ATR-FTIR (Thermo Fisher Scientific, Waltham, MA, USA) in a scanning range of 500–4000 cm^−1^.

### 3.5. Antibacterial Property Test

The antibacterial activity of SMCV was quantitatively determined using the shaking bottle method according to GB/T 20944.3-2008, GB/T 24346-2009, and AATCC 100-2004 as shown in Figure 7, using *E. coli* as test bacteria. The modified shake-flask method is mainly used to measure the antibacterial properties of antibacterial nanofibers. The antibacterial substances in the fiber membrane can fully come into contact with bacteria in the process of shaking. Compared with other antibacterial property testing methods, it has the characteristics of being quantitative, accurate, and objective. Activated *E. coli* was placed in sterilized Luria Bertani (LB) liquid medium and incubated at 37 °C for 17 h. The 30 mg of SMCV samples (SMCV-0.5, SMCV-1.0, SMCV-1.5, SMCV-2.0, SMCV-2.5, SMCV-3.0) were sterilized with UV positive and negative for 1 h before the experiment and the glassware was autoclaved at 121 °C for 20 min. The sterilized SMCV samples were added to a sterilized culture flask containing 30 mL bacterial suspension (0.05 × 10^8^ CFU/mL) and incubated at 37 °C for 24 h by shock. Pure *E. coli* suspensions were used as a control group. After 24 h, the *E. coli* suspension was continuously diluted with normal saline (0.85%); then, 200 μL of the diluted *E. coli* suspension was coated on LB solid medium with triangle coating stick. The AGAR plates were incubated at 37 °C for 24 h. All experiments were performed three times. The number of *E.coli* colonies in LB solid medium was counted by viable microbial counting method, and the antibacterial rate (*R*%) was calculated with Formula (1) [44].
(1)$$R=\left[\frac{X-Y}{X}\right]\times 100$$
where *R* refers to the antibacterial rate (%); *X* refers to the number of *E. coli* in the colony (the control); and *Y* refers to the number of *E. coli* colony SMCV.

## 4. Conclusions

In this work, Sc_2_O_3_-MgO particles were prepared and added into PCL/PVP electrospun nanofiber membranes. When 2.0 wt% Sc_2_O_3_-MgO was added, the membrane was composed of smooth fibers and loaded with white nanoparticles that had an average diameter of 256.2 nm. After 24 h, it showed good bacteriostatic performance against *E. coli* with a bacteriostatic rate of 100%. These results indicate that the prepared electrospun nanofiber membranes with good antibacterial properties can be used as the material needed for tissue engineering to prevent bacteria from multiplying in the injured site and solve the problem of postoperative infection.

## Figures and Tables

**Figure 1 ijms-24-07649-f001:**
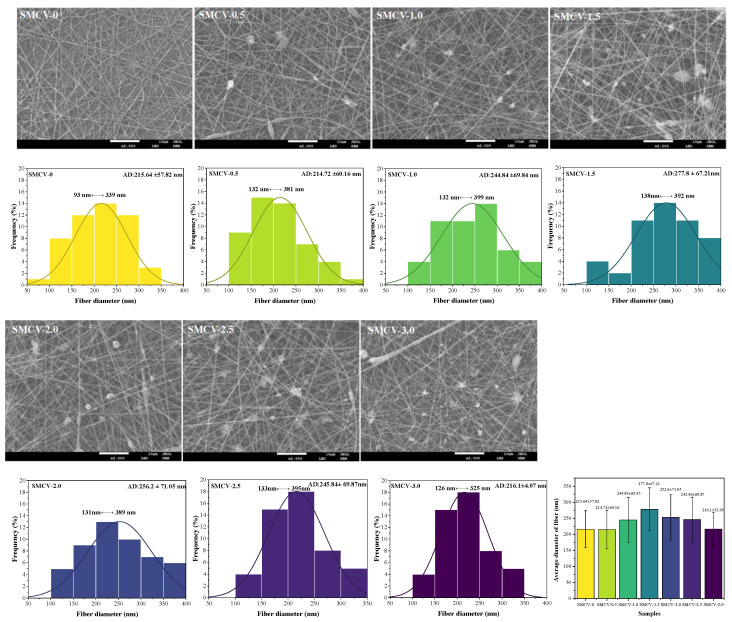
The SEM images, diameter distribution, and average diameters of SMCV (SMCV-0, SMCV-0.5, SMCV-1.0, SMCV-1.5, SMCV-2.0, SMCV-2.5, and SMCV-3.0).

**Figure 2 ijms-24-07649-f002:**
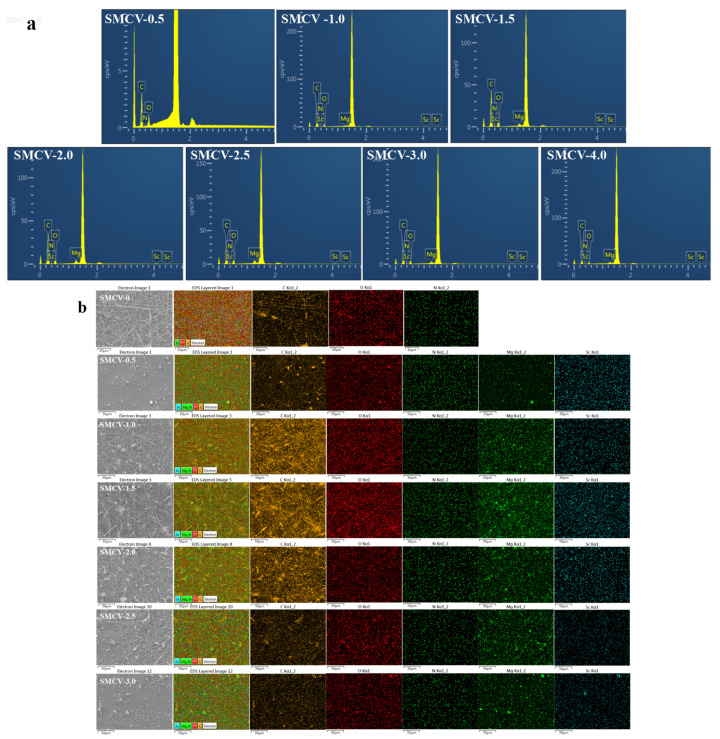
The elements analysis: (**a**) EDS spectrum of SMCV (SMCV-0, SMCV-0.5, SMCV-1.0, SMCV-1.5, SMCV-2.0, SMCV-2.5, and SMCV-3.0); (**b**) The elemental mapping of SMCV (SMCV-0, SMCV-0.5, SMCV-1.0, SMCV-1.5, SMCV-2.0, SMCV-2.5, and SMCV-3.0).

**Figure 3 ijms-24-07649-f003:**
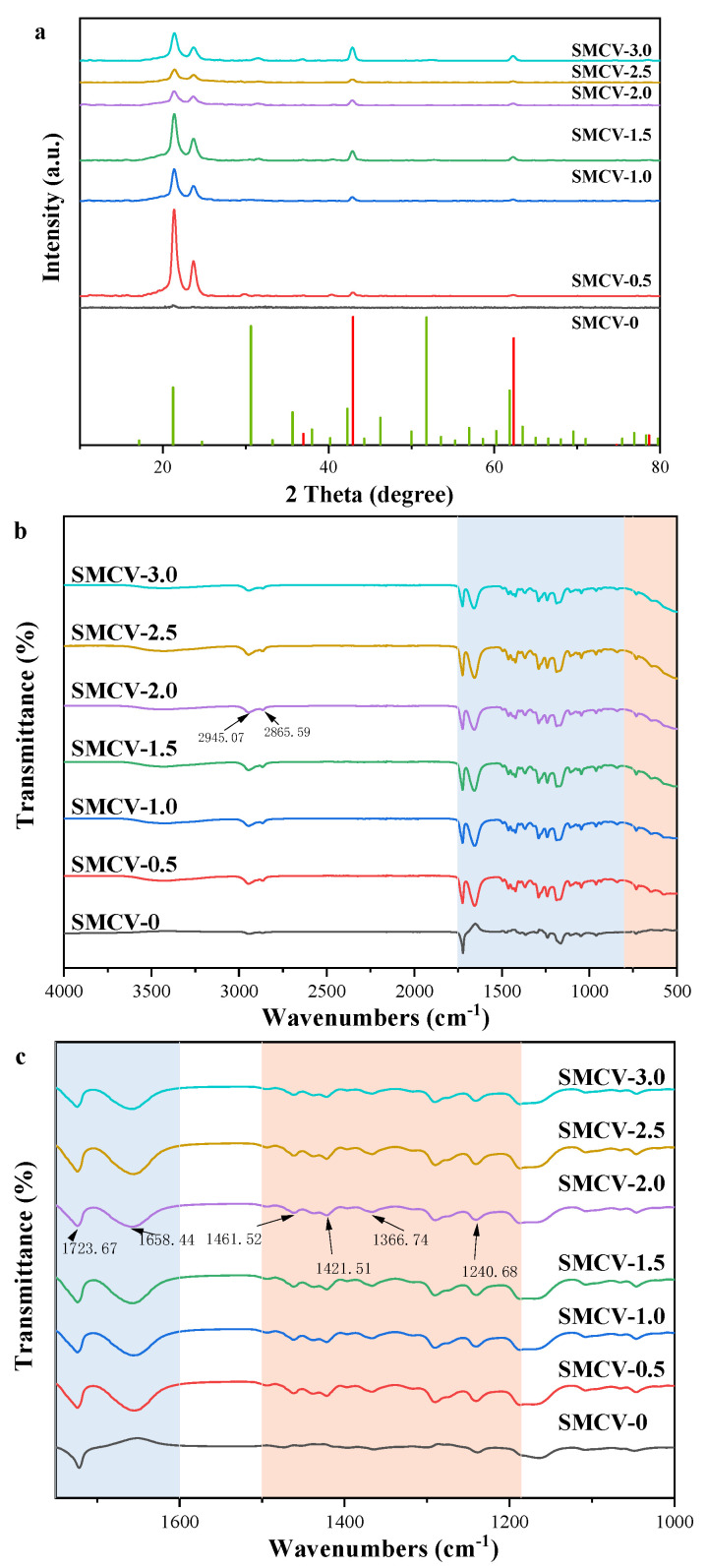
(**a**) XRD pattern of SMCV (SMCV−0, SMCV−0.5, SMCV−1.0, SMCV−1.5, SMCV−2.0, SMCV−2.5, and SMCV−3.0); (**b**) ATR spectra of SMCV (SMCV−0, SMCV−0.5, SMCV-1.0, SMCV−1.5. SMCV−2.0, SMCV−2.5, and SMCV−3.0).

**Figure 4 ijms-24-07649-f004:**
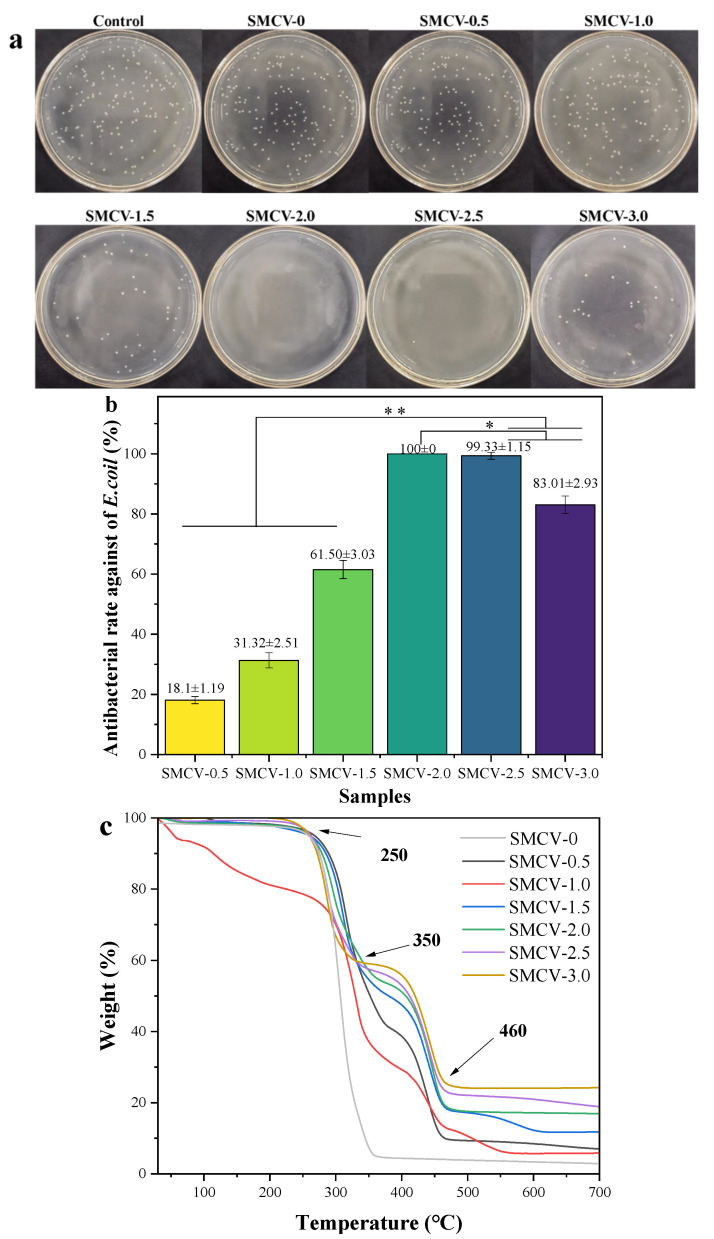
(**a**) The *E. coli* colony on LB solid medium (SMCV-0.5, SMCV-1.0, SMCV-1.5, SMCV-2.0, SMCV-2.5, and SMCV-3.0); (**b**) antibacterial rate of SMCV against *E. coli*, error bars represent mean ± SD for *n* = 3, * *p* < 0.05, ** *p* < 0.01; (**c**) TG curve of SMCV (SMCV-0, SMCV-0.5, SMCV-1.0, SMCV-1.5, SMCV-2.0, SMCV-2.5, and SMCV-3.0).

**Figure 5 ijms-24-07649-f005:**
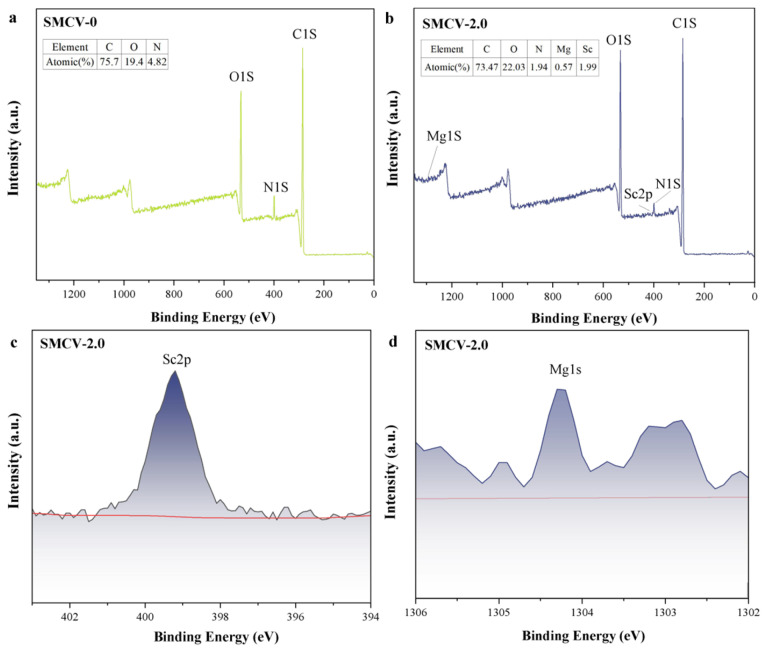
(**a**) XPS spectra of SMCV-0; (**b**) XPS spectra of SMCV-2.0; (**c**) Sc 2p XPS data of SMCV-2.0; (**d**) Mg 1s XPS data.

**Figure 6 ijms-24-07649-f006:**
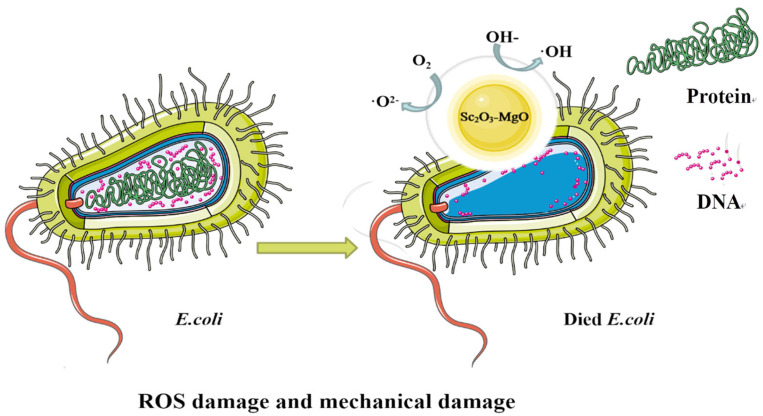
Antibacterial mechanism of Sc_2_O_3_-MgO.

**Figure 7 ijms-24-07649-f007:**
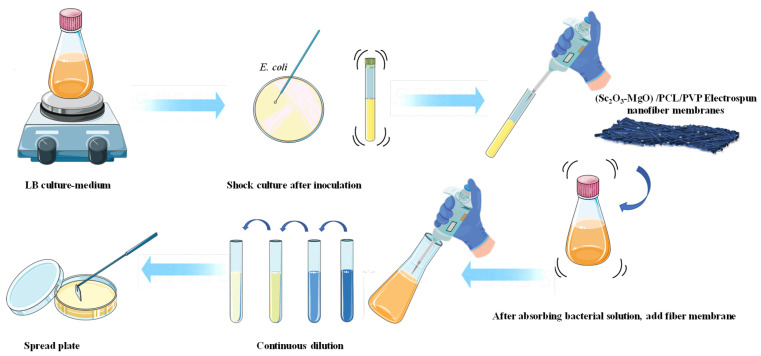
Operation steps of antibacterial experiment.

**Table 1 ijms-24-07649-t001:** The EDS data results of SMCV.

Index	Element	0	0.5	1.0	1.5	2.0	2.5	3.0
Sample labels		SMCV-0	SMCV-0.5	SMCV-1.0	SMCV-1.5	SMCV-2.0	SMCV-2.5	SMCV-3.0
	C		4.33	14.37	12.25	9.00	4.55	2.40
Apparent	N	3.54	1.11	2.41	2.25	1.77	1.13	0.78
Concentration	O	0.71	2.18	5.49	4.99	4.36	2.84	1.80
	Mg	1.47	0.23	0.82	1.14	1.14	0.78	0.58
	Sc		0.02	0.15	0.26	0.23	0.14	0.12
	C		0.04329	0.14375	0.12248	0.09005	0.04552	0.02402
	N	0.03536	0.00197	0.00430	0.00401	0.00315	0.00201	0.00138
K Ratio	O	0.00127	0.00732	0.01849	0.01679	0.01467	0.00955	0.00604
	Mg	0.00494	0.00150	0.00543	0.00754	0.00753	0.00516	0.00384
	Sc		0.00025	0.00152	0.00255	0.00234	0.00142	0.00120
	C		53.63	58.94	57.74	55.76	52.05	48.47
	N	52.99	12.76	10.41	10.40	9.87	10.21	10.81
Wt%	O	12.69	31.39	27.79	27.39	28.88	31.30	32.53
	Mg	34.32	1.98	2.36	3.56	4.45	5.34	6.65
	Sc		0.24	0.50	0.91	1.04	1.10	1.54
	C	1.18	0.57	0.45	0.35	0.53	0.72	0.83
Wt% Sigma	N	1.63	0.73	0.54	0.42	0.66	0.93	1.10
	O	0.96	0.45	0.33	0.26	0.41	0.59	0.71
	Mg		0.11	0.07	0.06	0.10	0.17	0.24
	Sc		0.11	0.07	0.06	0.10	0.16	0.22
	C		60.14	65.47	64.72	63.09	59.66	56.45
	N	59.12	12.27	9.92	10.00	9.57	10.04	10.79
Atomic%	O	12.14	26.43	23.17	23.04	24.53	26.94	28.44
	Mg	28.74	1.10	1.30	1.97	2.48	3.02	3.83
	Sc		0.07	0.15	0.27	0.31	0.34	0.48

**Table 2 ijms-24-07649-t002:** The *E. coli* colony number on LB solid medium of SMCV.

Samples	Control	SMCV-0	SMCV-0.5	SMCV-1.0	SMCV-1.5	SMCV-2.0	SMCV-2.5	SMCV-3.0
Colony number	182.0 ± 14.1	173.0 ± 14.5	141.6 ± 8.1	136.0 ± 9.2	45.3 ± 10.5	0 ± 0	1.0 ± 1.2	27.5 ± 8.1

**Table 3 ijms-24-07649-t003:** The sample labels of electrospun fibers.

Sc_2_O_3_-MgO Content (wt%)	0	0.5	1.0	1.5	2.0	2.5	3.0
Sample labels	SMCV-0	SMCV-0.5	SMCV-1.0	SMCV-1.5	SMCV-2.0	SMCV-2.5	SMCV-3.0

## Data Availability

Data openly available in a public repository.

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
