# Peer review of "Preparation and Properties of (Sc2O3-MgO)/Pcl/Pvp Electrospun Nanofiber Membranes for the Inhibition of Escherichia coli Infections"

_ijms, 2023, doi:10.3390/ijms24087649_

Round 1

Reviewer 1 Report

Dear authors 

The submitted manuscript entitled "Preparation and Properties of (Sc2O3-MgO)/Pcl/Pvp Electrospun Nanofiber Membranes for the Inhibition of Escherichia Coli Infections" contains significantly important data. However, some concerns were raised and needs to be addressed 

1- Line 131, correct the verbs used

2- The antibacterial effect was tested against only one microbe which seems insufficient to report the activity 

3- Line 147: absorbed and evenly coated in LB solid; seems misleading 

4- A growth curve of the tested bacteria must be added in presence and absence of the prepared nanofibers 

5- Control nanofibers must be tested 

6- Control Sc2O3-MgO alone must be characterized and  tested for possible antibacterial effect

7- The title Antibacterial mechanism analysis of SMCV was completely theoretically hence it can be moved to discussion or the author practically test the possible mechanism in vitro

Author Response

Response to Reviewers

Manuscript No.: ijms-2275611  

Title: Preparation and properties of (Sc2O3-MgO)/PCL/PVP electrospun nanofiber membranes for the inhibition of Escherichia coli infections

Journal: International Journal of Molecular Sciences

Thank you very much for your letter and for the reviewers’ comments concerning our manuscript entitled “Preparation and properties of (Sc2O3-MgO)/PCL/PVP electrospun nanofiber membranes for the inhibition of Escherichia coli infections”. The comments were all valuable and very helpful for revising and improving our paper, and the important suggestions were significant to our researches. We have studied the comments carefully and have made corrections which we hope meet with approval. The revised portions were marked in red in the revised manuscript. The main corrections in the paper and the response to the reviewer’s comments are attached as follows.

Looking forward to your positive response.

Sincerely yours,

Ying Wang

[email protected]

For reviewer 1:

The submitted manuscript entitled "Preparation and Properties of (Sc2O3-MgO)/PCL/PVP Electrospun Nanofiber Membranes for the Inhibition of Escherichia Coli Infections" contains significantly important data. However, some concerns were raised and needs to be addressed.

1.Line 131, correct the verbs used.

Thank you very much for your careful reading of our manuscript. The verbs was corrected in our work.

2.The antibacterial effect was tested against only one microbe which seems insufficient to report the activity. 

Thank you very much for your significant comment and suggestion. As you said, it would be better if we do the antibacterial test against another bacteria. However, we cannot complete this experiment within ten days. Your significant valuable comments benefit us a lot, and the process of ultrasonic dispersion will be added in our next work.

3.Line 147: absorbed and evenly coated in LB solid; seems misleading. 

Thank you very much for your valuable suggestion. We have modified this content, and the revised manuscript is shown as below:

(Line 185-187, page 4)

After 24 h, the E. coli suspension was continuously diluted with normal saline (0.85%), then 200 μL of the diluted E. coli suspension was coated on LB solid medium with triangle coating stick.

4.A growth curve of the tested bacteria must be added in presence and absence of the prepared nanofibers. 

Thank you very much for your valuable advice, In this work, we focus on the preparation and antibacterial activity of fiber membranes. And in our next research entitled “Study on the antibacterial mechanism of (Sc2O3-MgO)/PCL/PVP electrospun nanofiber membranes against Escherichia coli model”, we focused on the antibacterial mechanism of fiber membranes against bacterial model, including the inhibition of bacterial growth curves as you said. Thanks again for your thoughtful suggestion.

5.Control nanofibers must be tested. 

Thank you very much for your significant suggestion. The control nanofibers have been tested in our research. The revised manuscript is shown as below:

(Line 266-267, page 8)

3.3. XRD

After loading for Sc2O3-MgO, the nanofibers have sharper diffraction peaks, indicating that the crystallinity is improved after loading.

(Line 280-281, page 8)

3.4. ATR spectra analysis

Compared with SMCV-0, bands at 3695cm−1 showed the presence of Sc2O3-MgO nanoparticles.

(Line 314-319, page 10)

3.5. Thermal analysis of nanofibers

SMCV-0 starts to lose weight at 250oC, and loses weight completely at 355oC. Compared with SMCV-0, the thermogravimetric temperature of SMCV after loading Sc2O3-MgO is delayed by 10 ℃. As a result, the thermal stability of SMCV-0 is significantly higher than that of original SMCV-0, and the residual mass is also higher than that of unloaded nanofiber.

6.Control Sc2O3-MgO alone must be characterized and tested for possible antibacterial effect. 

Thanks for your valuable and thoughtful evaluation. We have characterized and tested the  Sc2O3-MgO in our recent work (Research on synthesis and property of nano‑textured Sc2O3‑MgO effcient antibacterial agents, JBIC Journal of Biological Inorganic Chemistry, 2023).

7.The title Antibacterial mechanism analysis of SMCV was completely theoretically hence it can be moved to discussion or the author practically test the possible mechanism in vitro.

Thank you very much for the warning, we have moved the “Antibacterial mechanism analysis of SMCV” to the “discussion”, which was marked in red in the revised manuscript .

(Line 381-400, page 11)

There are several theories about the antibacterial mechanism of MgO, but the production of reactive oxygen species (ROS) and mechanical damage is the main mechanisms. on the one hand relying on the morphological structure of magnesium oxide itself to physically damage the membrane system of bacteria, leading to cell rupture and death; on the other hand, the superoxide ions generated on its surface, and the alkaline environment caused by the reaction with water, can cause the denaturation of bacterial DNA and proteins[29-31]. As shown in Fig. 6, Sc2O3-MgO has strong antibacterial activity. One possible mechanism for its bacterial virulence includes the release of O2- ,hydroxyl radical OH and severe damage to cellular components by reactive oxygen species (ROS)[28]. These free radicals can damage cell structure through strong REDOX properties, and prevent the normal reproduction of bacteria, so as to achieve antibacterial effect. On the other hand, there are many kinds of active sites on the surface of Sc2O3-MgO, such as lattice limited hydroxyl, free hydroxyl and ion holes, which can be used as adsorption and surface reaction centers and cause mechanical damage. Scandium (Sc3+) has been shown to have the ability to inhibit the growth of Klebsiella pneumoniae, E. coli and Pseudomonas aeruginosa[32-34].The purpose of doping Sc3+ in the nano-MgO lattice is to increase the lattice defects and then increase the ROS content produced during the antibacterial process to enhance the antibacterial property of MgO. When Sc2O3-MgO is successfully loaded on PCL/PVP, it has good antibacterial performance.

Reviewer 2 Report

The study aimed to develop an efficient antibacterial nanofiber membrane for tissue engineering using electrospun nanofiber membranes loaded with Sc2O3-MgO synthesized by doping Sc3+ and calcining at 600°C. The morphology of all formulations was studied using SEM and EDS, and further analysis was done using XRD, TGA, and ATR-FTIR. The results showed that the PCL/PVP nanofibers loaded with 2.0 wt% Sc2O3-MgO were smooth and homogeneous with an average diameter of 252.6 nm, and a low load concentration of 2.0 wt% Sc2O3-MgO showed a 100% antibacterial rate against Escherichia coli. The electrospun nanofiber membranes are promising for biomedical applications due to their high porosity, large specific surface area, and structural similarity with the extracellular matrix.

Couple of comments for the authors:

  • It would be helpful to provide more background information on the current state of antibacterial wound dressings, and how nanofiber membranes can improve upon existing solutions.
  • The introduction briefly mentions the advantages of electrospun nanofiber membranes, but it would be useful to expand on this and explain why they are particularly suitable for wound dressings.
  • While the introduction does a good job of describing the properties of PCL and PVP, it would be helpful to also explain why these materials were chosen for use in this study.
  • The introduction discusses the antibacterial properties of nano-MgO, but doesn't explain why its antibacterial performance is not ideal. It would be helpful to provide some context and background information here.
  • The authors mention the use of Sc2O3-MgO with nano-textured surface as antibacterial agents, but it is not clear how these agents were incorporated into the nanofiber membranes. Some additional information on this would be helpful.
  • It would be useful to provide more information on the modified shake-flask method used to test the antibacterial performance of SMCV against E. coli. How was this method chosen, and how does it compare to other testing methods?

  • Can the authors explain why they chose the specific range of Sc2O3-MgO loading percentages used in the study?
  • What is the significance of the observed increase in particulate matter on the nanofibers with increasing Sc2O3-MgO loading, and how does it relate to the antibacterial properties of the material?
  • Can the authors provide more information about the effects of Sc2O3-MgO loading on the mechanical properties of the nanofiber membranes, and how these properties may affect their potential applications in wound dressings or other medical devices?
  • Are there any potential limitations or drawbacks to using Sc2O3-MgO as an antibacterial agent in electrospun nanofibers, and how might these affect the overall effectiveness of the material for medical or other applications?
  • What further research or experimentation is needed to fully understand the effects of Sc2O3-MgO loading on electrospun nanofiber morphology, and how might these findings be applied in future studies or real-world applications?
  • Include a visible and easy to see scale bar in Figure 2, panel b.
  • Figure 5, panel b needs statistics (t-test, p values) to draw conclusions, as compared to the control. 

Author Response

Response to Reviewers

Manuscript No.: ijms-2275611  

Title: Preparation and properties of (Sc2O3-MgO)/PCL/PVP electrospun nanofiber membranes for the inhibition of Escherichia coli infections

Journal: International Journal of Molecular Sciences

Thank you very much for your letter and for the reviewers’ comments concerning our manuscript entitled “Preparation and properties of (Sc2O3-MgO)/PCL/PVP electrospun nanofiber membranes for the inhibition of Escherichia coli infections”. The comments were all valuable and very helpful for revising and improving our paper, and the important suggestions were significant to our researches. We have studied the comments carefully and have made corrections which we hope meet with approval. The revised portions were marked in red in the revised manuscript. The main corrections in the paper and the response to the reviewer’s comments are attached as follows.

Looking forward to your positive response.

Sincerely yours,

Ying Wang

[email protected]

For reviewer 2:

The study aimed to develop an efficient antibacterial nanofiber membrane for tissue engineering using electrospun nanofiber membranes loaded with Sc2O3-MgO synthesized by doping Sc3+ and calcining at 600°C. The morphology of all formulations was studied using SEM and EDS, and further analysis was done using XRD, TGA, and ATR-FTIR. The results showed that the PCL/PVP nanofibers loaded with 2.0 wt% Sc2O3-MgO were smooth and homogeneous with an average diameter of 252.6 nm, and a low load concentration of 2.0 wt% Sc2O3-MgO showed a 100% antibacterial rate against Escherichia coli. The electrospun nanofiber membranes are promising for biomedical applications due to their high porosity, large specific surface area, and structural similarity with the extracellular matrix.

Couple of comments for the authors:

1.It would be helpful to provide more background information on the current state of antibacterial wound dressings, and how nanofiber membranes can improve upon existing solutions.

Thank you very much for the valuable suggestion, some contents were added in the introduction. The revised manuscript is shown as below:

(Line 24-Line 31, page 1)

Wound dressing is a covering or protective layer that can temporarily protect the damaged skin in the process of wound healing and treatment and avoid or control wound infection[1]. Therefore, it is very important to develop wound dressing that can prevent bacteria from penetrating into the wound or avoid microbial growth. Over the past few decades, extensive research has been conducted on wound dressings with antibacterial properties, such as thin films, hydrogels, emulsions, composites, nano/microfibers, etc[2]. In recent years, nano/microfibers have shown broad application prospects in nano-wound dressing.

2.The introduction briefly mentions the advantages of electrospun nanofiber membranes, but it would be useful to expand on this and explain why they are particularly suitable for wound dressings. 

Thank you very much for your valuable advice, some contents were added in the introduction. The revised manuscript is shown as below:

(Line 31-Line 34, page 1)

The electrospun nanofiber structure possesses absorbability, bacterial barrier, oxygen permeability (gas transfer), non-adhesion to healing tissue, and biological activity, all of which are essential properties for antibacterial wound dressings[3].

3.While the introduction does a good job of describing the properties of PCL and PVP, it would be helpful to also explain why these materials were chosen for use in this study.

Thank you very much for the valuable suggestion, we have explained the issue as below:

(Line 54-Line 62, page 2)

PCL is characterized by good biocompatibility, non-toxicity, adjustable degradation rate, permeability with many drugs, and complete absorption and metabolism from the human body, which is widely used in controlled drug release systems as a sustained release carrier material [14]. It is commonly used in surgical sutures, fracture internal fixation devices, drug delivery, and tissue or organ regeneration scaffolds[15]. However, PCL is a hydrophobic material with strong crystallization and poor hydrophilicity, which hinders its application in medical fields where rapid absorption rate is needed [16]. Its slow degradation kinetics may hinder its application in some biomedical applications that require faster absorption rates.

(Line 64-Line 69, page 2)

Polyvinylpyrrolidone (PVP) is a biodegradable, biocompatible, water soluble, ph stable, non-toxic amphiphilic polymer with good solubility, viscosity and film forming performance with a variety of organic solvents[13]. PVP has good electrostatic spinning properties, including spinning and fiber extraction, and is widely used in the preparation of nanocapsules, implant materials and scaffolds.

4.The introduction discusses the antibacterial properties of nano-MgO, but doesn't explain why its antibacterial performance is not ideal. It would be helpful to provide some context and background information here.

Thanks for your valuable and thoughtful evaluation, there is a mistake in the expression of this sentence. The revised manuscript is shown as below:

(Line 81-Line 86, page 2)

Organic antimicrobials have strong initial bactericidal power, but their chemical stability is poor, and they are easy to volatilize when exposed to heat, light or water, so it is difficult to achieve long-lasting effect, and even produce toxic decomposition products.The smaller the particle size of nanometer magnesium oxide, the better the growth inhibition and destruction of bacteria. In addition, reactive oxygen species(ROS) play an important role in the antibacterial mechanism of magnesium oxide, This depends mainly on the oxygen vacancy and alkalinity of magnesium oxide. Ion doping is an effective method to modify the physical and chemical properties of metal oxides, such as small particle size, high defect concentration and high catalytic activity.

5.The authors mention the use of Sc2O3-MgO with nano-textured surface as antibacterial agents, but it is not clear how these agents were incorporated into the nanofiber membranes. Some additional information on this would be helpful.

Thank you very much for your valuable evaluations and suggestion. Some additional information was added to the introduction. The revised manuscript is shown as below:

(Line 100-Line 106, page 3)

By electrospinning technology, Sc2O3-MgO was mixed into spinning solution to prepare PCL/PVP nanofiber membrane, which had antibacterial properties. Electrospun fiber membranes loaded with bioactive materials can be used for wound dressing, drug release and artificial tissue engineering scaffolds. Loaded nanoparticles physically protect wounds from bacterial activity. Nanofibers can help cell differentiation and proliferation.

6.It would be useful to provide more information on the modified shake-flask method used to test the antibacterial performance of SMCV against E. coli. How was this method chosen, and how does it compare to other testing methods?

Thank you very much for your valuable and thoughtful evaluations. We have already revised the section of “Antibacterial property test”. The revised manuscript is shown as below:

(Line 175-Line 179, page 4)

The modified shake-flask method is mainly used to measure the antibacterial properties of antibacterial nanofibers. The antibacterial substances in the fiber membrane can fully contact with bacteria in the process of shaking. Compared with other antibacterial properties testing methods, it has the characteristics of quantitative, accurate and objective.

7.Can the authors explain why they chose the specific range of Sc2O3-MgO loading percentages used in the study?

Thank you very much for your valuable evaluation, we have explained the issue as below:

In our previous research, the PCL/PVP nanofibers loaded with nano-MgO and nano-textured nano-MgO were prepared through electrospinning technology. When the nano-MgO reached to 0.25 wt.%, 0.5 wt.%, 1.0 wt.%, 3.0 wt.% and 5.0 wt.%, the antibacterial ability against E. coli can reach 27.2%, 31.5%, 82.8%, 100% and 100%, respectively. And when the nano-textured nano-MgO reached to 0.50 wt.%, 1.0 wt.%, 2.0 wt.%, 2.5 wt.%, 3.0 wt.% and 5.0 wt.%, the antibacterial ability against E. coli can reach 7.3%, 24.3%, 64.4%, 73.1%, 100.0%, 100.0%, respectively. Thus, the specific range of Sc2O3-MgO loading percentages were used in this study.

8.What is the significance of the observed increase in particulate matter on the nanofibers with increasing Sc2O3-MgO loading, and how does it relate to the antibacterial properties of the material?

Thank you very much for your evaluation and suggestion. We have explained the issue as below:

Within a certain range, the antibacterial property of the material enhanced with the increasing of Sc2O3-MgO loading. However, when the loaded antibacterial agent is excessive, the powders can cause serious aggregation within the nanofibers, which can seriously reduce the antibacterial property of the material.

9.Can the authors provide more information about the effects of Sc2O3-MgO loading on the mechanical properties of the nanofiber membranes, and how these properties may affect their potential applications in wound dressings or other medical devices?

Thanks again for your valuable suggestion, we have explained the issue as below:

In view of our previous research, due to the amount of the Sc2O3-MgO powders loaded in the nanofibers is less, the mechanical properties of the fiber membrane after loading the powders have no obvious difference from that before loading.

10.Are there any potential limitations or drawbacks to using Sc2O3-MgO as an antibacterial agent in electrospun nanofibers, and how might these affect the overall effectiveness of the material for medical or other applications?

Thank you very much for your valuable evaluation, we have explained the issue as below:

The SMCV prepared by electrospinning technology has good biodegradability and is in line with the anisotropy of the fiber ring, which is conducive to cell adhesion and growth. However, excessive load of antibacterial agent Sc2O3-MgO will form the agglomeration state, leading to pore obstruction and affecting cell proliferation.

11.What further research or experimentation is needed to fully understand the effects of Sc2O3-MgO loading on electrospun nanofiber morphology, and how might these findings be applied in future studies or real-world applications?

Thank you very much for your significant comments and suggestions. As you said, further research or experimentation is needed to fully understand the effects of Sc2O3-MgO loading on electrospun nanofiber morphology. Your significant valuable comments benefit us a lot, and the process of ultrasonic dispersion will be added in our next work. Thanks again for your advice. 

12.Include a visible and easy to see scale bar in Figure 2, panel b. 

Thanks for your warning, the a visible and easy to see scale bar have been added in Figure 2. The revised manuscript is shown as below:

(Line 223-225, page 6)

Figure 2. The SEM images, diameter distribution and average diameters of SMCV (SMCV-0, SMCV-0.5, SMCV-1.0, SMCV-1.5, SMCV-2.0, SMCV-2.5 and SMCV-3.0).

13.Figure 5, panel b needs statistics (t-test, p values) to draw conclusions, as compared to the control. 

Thanks again for your valuable suggestion, the error bar and statistics (t-test, p values) was added in the Figure 5. The revised manuscript is shown as below:

(Line 301, page 10)

Figure 5. b. antibacterial rate of SMCV against E. coli, Error bars represent mean ± SD for n = 3, * p < 0.05, **p < 0.01; c. TG curve of SMCV(SMCV-0, SMCV-0.5, SMCV-1.0, SMCV-1.5, SMCV-2.0, SMCV-2.5 and SMCV-3.0 ).

(Line 349-352, page 12)

As shown in Figure 6b, SMCV has certain antibacterial ability against E. coli, and the antibacterial rate increases significantly with the increase of loading capacity, while the antibacterial performance weakens when the load is too large (18.1±1.19, 31.32±2.52, 61.50±3.03, 68.88±1.64, 100±0, 84.68±4.07%, p < 0.05 or p < 0.01).

Reviewer 3 Report

I think the manuscript would be worth of publishing in the Journal providing following remarks/suggestions will be accounted for. I hope that they may help improving the manuscript appropriately. Alternatively, the authors should give reasonable comments and answers.

1.It is not necessary to include all of your data in a single article. Also use statistics appropriately, and be clear on the need to justify the sophistication of analysis in the context of the validity of underlying data. 

2.I congratulate you on producing a paper that, , is for the most part very clear and understandable. I very much appreciate the effort that this requires.

3.I would like however to see more details about what is the reaction mechanisms ( synthesize nano-structured Sc2O3-MgO by doping Sc3+ ,  loaded it onto the PCL/PVP substrates by electrospinning technology)are incorporated in the manuscript, some comments about the potential application in a biomaterial.

4.Authors can consider doing this XPS analysis

5.What is the interest of the surface chemical composition on  on FTIR ? Are they correlated to a better or worse antibacterial rate? 

6. The biocompatibility (ex: cytotoxicity, in vitro experiment) reference of the  (Sc2O3-MgO)/PCL/PVP electrospun nanofiber membranes must be added.

7.The author should write the purpose for each test in one/two sentences (in brief) before explaining the results of the characterization techniques. 

8.Figure 2 should add  the error value.

9.Does  have the zone  of inhibition the nanofiber membranes?

Author Response

Response to Reviewers

Manuscript No.: ijms-2275611  

Title: Preparation and properties of (Sc2O3-MgO)/PCL/PVP electrospun nanofiber membranes for the inhibition of Escherichia coli infections

Journal: International Journal of Molecular Sciences

Thank you very much for your letter and for the reviewers’ comments concerning our manuscript entitled “Preparation and properties of (Sc2O3-MgO)/PCL/PVP electrospun nanofiber membranes for the inhibition of Escherichia coli infections”. The comments were all valuable and very helpful for revising and improving our paper, and the important suggestions were significant to our researches. We have studied the comments carefully and have made corrections which we hope meet with approval. The revised portions were marked in red in the revised manuscript. The main corrections in the paper and the response to the reviewer’s comments are attached as follows.

Looking forward to your positive response.

Sincerely yours,

Ying Wang

[email protected]

For reviewer 3:

I think the manuscript would be worth of publishing in the Journal providing following remarks/suggestions will be accounted for. I hope that they may help improving the manuscript appropriately. Alternatively, the authors should give reasonable comments and answers.

1.It is not necessary to include all of your data in a single article. Also use statistics appropriately, and be clear on the need to justify the sophistication of analysis in the context of the validity of underlying data.

Thank you very much for your valuable evaluations and suggestions. We will improve the data in our work.

2.I congratulate you on producing a paper that, , is for the most part very clear and understandable. I very much appreciate the effort that this requires.

Thank you for your encouragement. We will work harder and create more achievements in this field.

3.I would like however to see more details about what is the reaction mechanisms ( synthesize nano-structured Sc2O3-MgO by doping Sc3+, loaded it onto the PCL/PVP substrates by electrospinning technology)are incorporated in the manuscript, some comments about the potential application in a biomaterial.

Thank you very much for your careful reading of our manuscript, we have revise the reaction mechanisms in our research. The revised manuscript is shown as below:

(Line 100-105, page 3)

By electrospinning technology, Sc2O3-MgO was mixed into spinning solution to prepare PCL/PVP nanofiber membrane, which had antibacterial properties. Electrospun fiber membranes loaded with bioactive materials can be used for wound dressing, drug release and artificial tissue engineering scaffolds. Loaded nanoparticles physically protect wounds from bacterial activity. Nanofibers can help cell differentiation and proliferation.

4.Authors can consider doing this XPS analysis.

Thank you very much for your valuable and thoughtful suggestion. The XPS analysis was considered in this work. The revised manuscript is shown as below:

(Line 329-340, page 11)

Figure 6. a. XPS spectra of SMCV-0 ; b. XPS spectra of  SMCV-2.0 ;c. Sc 2p XPS data of SMCV-2.0; d. Mg 1s XPS data.

3.6. XPS analysis of nanofibers

XPS was used to test SMCV nanofiber membrane before and after loading Sc2O3-MgO, and the content changes of elements on the surface of the membrane were determined. Before and after loading Sc2O3-MgO on SMCV nanofiber membrane with XPS, the content of elements on the membrane surface was measured. Figure 6a,b is the full-scan spectrum analysis of SMCV-0 and SMCV-2.0 surface element content, showing that the test samples mainly contain characteristic peaks of C, O and N. In the figure, there is no impurity peak of other substances, and the content of C element and O element in the nanofiber membrane is above 95%, which is the main component of PCL/PVP spinning substrate. As shown in Figure 6 c and d, the increases of 1303.3eV and 398.4eV are mainly attributed to the characteristic peaks of Mg2p and Sc2p, the doping of Sc3+ may change the molecular structure of MgO, in which the increase of Sc3+ is more obvious, indicating that Sc2O3-MgO is successfully loaded on SMCV.

5.What is the interest of the surface chemical compositionon FTIR? Are they correlated to a better or worse antibacterial rate?  

Thanks again for your valuable and thoughtful evaluation. We explain this issue as below:

ATR was used to further verify the chemical composition of SMCV nanofiber membrane. It was used to detect whether new chemical bonds and molecular interactions were generated in SMCV nanofiber, and to explore whether the chemical bonds of the nanofiber membrane changed after loading antibacterial agents.

  1. The biocompatibility (ex: cytotoxicity, in vitro experiment) reference of the  (Sc2O3-MgO)/PCL/PVP electrospun nanofiber membranes must be added.

Thank you very much for your significant comments and suggestions. As you said, it would be better if the biocompatibility reference of the (Sc2O3-MgO)/PCL/PVP electrospun nanofiber membranes are added in our article. However, we can not complete the experiments within 10 d. Your significant valuable comments benefit us a lot, and the process of ultrasonic dispersion will be added in our next work. Thanks again for your advice.

7.The author should write the purpose for each test in one/two sentences (in brief) before explaining the results of the characterization techniques.

Thank you very much for your suggestion, one/two sentences (in brief) before explaining the results of the characterization techniques were added in our article as you said. The revised manuscript is shown as below:

(Line 198-201, page 5)

3.1. Morphology and diameter analysis

In order to simulate the structure and function of ECM, the tissue engineering structure must be conducive to promoting cell adhesion and proliferation. Scanning electron microscopy was used to observe the structure of nanofiber scaffolds, and the influence of the change of Sc2O3-MgO load on the morphology of nanofibers was studied.

(Line 230-232, page 6)

3.2. EDS spectrum

EDS characterization was used to further determine the loading of Sc2O3-MgO and the distribution of elements in PCL/PVP nanofiber membranes, and the types and contents of SMCV components were analyzed.

(Line 259-260, page 8)

3.3. XRD

The crystal state of SMCV loaded with antibacterial agent was analyzed by XRD test, and the corresponding diffraction curve was analyzed and studied.

(Line 275-277, page 8)

3.4. ATR spectra analysis

ATR was used to test whether new chemical bonds and molecular interactions were generated in SMCV nanofibers, and the functional groups of organic compounds were quickly and effectively identified.

(Line 307-308, page 10)

3.5. Thermal analysis of nanofibers

Thermogravimetric analyzer (TG) was used to test the thermodynamic properties of SMCV nanofibers and characterize their thermal stability.

(Line 329-331, page 11)

3.6. XPS analysis of nanofibers

XPS was used to test SMCV nanofiber membrane before and after loading Sc2O3-MgO, and the content changes of elements on the surface of the membrane were determined. 

8.Figure 2 should add the error value. 

Thank you very much for your careful reading of our manuscript, we were so sorry for this mistake. We have cited the mistake, and the revised manuscript is shown as below:

(Line 225, page 6)

Figure 2. The SEM images, diameter distribution and average diameters of SMCV (SMCV-0, SMCV-0.5, SMCV-1.0, SMCV-1.5, SMCV-2.0, SMCV-2.5 and SMCV-3.0).

9.Does have the zone of inhibition the nanofiber membranes?

Thank you very much for your valuable and thoughtful evaluation. We explain this issue as below:

The Sc2O3-MgO does not belong to a dissolution type antibacterial mechanism, so that the zone of inhibition of the nanofiber membranes is not obvious and cannot reveal the antibacterial effect of different nanofiber membranes.

Round 2

Reviewer 1 Report

Thank you for considering the comments 

Author Response

Thank you very much for your comments on this article.